# Defects Recognition Algorithm Development from Visual UAV Inspections

**DOI:** 10.3390/s22134682

**Published:** 2022-06-21

**Authors:** Nicolas P. Avdelidis, Antonios Tsourdos, Pasquale Lafiosca, Richard Plaster, Anna Plaster, Mark Droznika

**Affiliations:** 1School of Aerospace, Transport & Manufacturing, Cranfield University, Cranfield MK43 0AL, UK; a.tsourdos@cranfield.ac.uk (A.T.); pasquale.lafiosca@cranfield.ac.uk (P.L.); 2ADDIT, 17 Railton Road, Wolseley Business Park, Kempston, Bedford MK42 7PN, UK; richardjamesplaster@gmail.com (R.P.); annaplaster@gmail.com (A.P.); 3TUI Airline, Area 8, Hangar 61, Percival Way, London Luton Airport, Luton LU2 9PA, UK; mark.droznika@tui.co.uk

**Keywords:** defect recognition, aircraft inspection, deep learning, CNN, UAV, defect classification, AI

## Abstract

Aircraft maintenance plays a key role in the safety of air transport. One of its most significant procedures is the visual inspection of the aircraft skin for defects. This is mainly carried out manually and involves a high skilled human walking around the aircraft. It is very time consuming, costly, stressful and the outcome heavily depends on the skills of the inspector. In this paper, we propose a two-step process for automating the defect recognition and classification from visual images. The visual inspection can be carried out with the use of an unmanned aerial vehicle (UAV) carrying an image sensor to fully automate the procedure and eliminate any human error. With our proposed method in the first step, we perform the crucial part of recognizing the defect. If a defect is found, the image is fed to an ensemble of classifiers for identifying the type. The classifiers are a combination of different pretrained convolution neural network (CNN) models, which we retrained to fit our problem. For achieving our goal, we created our own dataset with defect images captured from aircrafts during inspection in TUI’s maintenance hangar. The images were preprocessed and used to train different pretrained CNNs with the use of transfer learning. We performed an initial training of 40 different CNN architectures to choose the ones that best fitted our dataset. Then, we chose the best four for fine tuning and further testing. For the first step of defect recognition, the DenseNet201 CNN architecture performed better, with an overall accuracy of 81.82%. For the second step for the defect classification, an ensemble of different CNN models was used. The results show that even with a very small dataset, we can reach an accuracy of around 82% in the defect recognition and even 100% for the classification of the categories of missing or damaged exterior paint and primer and dents.

## 1. Introduction

Air transport is one of the most significant ways of moving people across the globe. In 2019, the number of air passengers carried worldwide was around 4.2 billion, an overall increase of 92% compared with 2019 [1]. During COVID-19, most travelling was put almost on a halt with the numbers decreasing significantly. In 2020, the total number of passengers dropped significantly to around one billion (1034 million) [2]. As a result, the need of reducing costs across the industry has become imminent. Around 10–15% of the operational costs of an airline are around maintenance, repairs, and overhaul (MRO) activities [3]. Currently, aircraft maintenance heavily involves visual tasks carried by humans [3]. This is very time consuming, costly and introduces possibilities for human errors. It is understood that automating these visual tasks could solve this problem [4,5,6]. For this reason, the use of climbing robots or UAVs to perform these tasks have been attempted. Climbing robots usually use magnetic forces, suction caps, or vortexes to climb to the aircraft structure [7,8,9]. However, robotic platforms for inspection face difficulties in achieving good adherence and mobility due to their lack of flexibility [7,10,11]. On the other hand, UAVs have been proposed for the inspection [12,13,14,15] of buildings, wind turbines, power transmission lines and aircrafts. UAVs could minimize inspection time and cost as they can inspect quickly large areas of the aircraft and data can be transmitted to a ground base in real time for processing. The key challenge of all the above automated techniques is developing defect detection algorithms that are able to perform with accuracy and repeatability. Several attempts have been made and most of them can be divided into the following two categories: the ones that use more traditional image processing techniques [5,16,17,18] and the ones that use machine learning [19,20,21,22,23,24]. In the first category, image features such as convexity or signal intensity [5] are used. In [18], the authors proposed a method using histogram comparisons or structural similarity. In addition, in [16,17], the authors proposed the use of neural networks trained on feature vectors extracted from contourlet transform. These techniques have very good accuracy in the test data but are failing to effectively generalize and need continuous tuning. On the other hand, algorithms using convolutional neural networks (CNN) have showed good results in defect detection [19,20,21,25]. In [19,20], CNNs are used as feature extractors and then either a single shot multibox detector (SSD) or a support vector machine (SVM) are used for the classification. The use of faster CNN is also proposed for classification and localization [22]. In addition, the use of UAVs together with deep learning algorithms is proposed for the tagging and localization of concrete cracks [26,27].

The main challenge of the machine learning algorithms is the requirement of a large amount of data. Especially for the CNNs, the amount of data required can be in the scale of thousands, especially if a model is not already pretrained. The existence of large datasets in concrete structures has allowed CNNs to show excellent results in defect detection in concrete structures. On the other hand, in aircraft structures, the results are promising but are still not very accurate [18] or they deal with only the problem of defect recognition [20]. In this paper, we propose a two-step classification process of an ensemble of machine learning classifiers for both defect recognition and classification. In this two-step process, we are using pretrained CNNs to both recognize and classify a series of defects in aircraft metallic and composite structures. In the first step, we are performing the defect recognition and in the second step, the defect classification. By combining the results of different classifiers, we can more effectively address the issue of small datasets and produce results with an accuracy reaching 82% in the defect recognition step.

## 2. Dataset

One of the challenges in this study was the creation of datasets for training and testing the classifiers. As most of the datasets of defects on aircrafts are not public available, we needed to create our own. The datasets were created with the help and permission of TUI© [28]. The images were taken during the scheduled maintenance of aircrafts in TUI’s base maintenance hangar in Luton, UK. The imaging sensor used was a SONY RX0 II© rugged mini camera. This model can be carried by a drone and is able to take images from any angle. All the technical specifications of the camera, such as sensor size and type, focal length, size of the sensor, are widely available and the effective resolution is 15-megapixels with maximum resolution of 4800 × 3200 pixels. Images for the datasets were captured and the following seven types of defects were investigated:Missing or damaged exterior paint and primer;Dents;Lighting strike damage;Lighting strike fastener repair;Reinforcing ratch repairs;Nicks, scratches, and gouges;Blend/rework repairs.

In Figure 1, images of the defects are presented. Most of the obtained images contained several defects, together with other elements such as screws, etc. In order to create the two different datasets, further processing was needed to extract only the objects that we were interested in from each of the images.

The objects of interest were cropped through a semi-automated procedure to create the datasets for the training. A Python script was developed so the user can select and crop the area with the object of interest. The cropped image was saved in the new image file. The name of the file was indicative of the category of the defect. This provided us the capability to extract multiple images of interest from only one image, with and without defects. The cropped images were grayscaled because we did not want the classifiers to associate color with any defects during training. This was carried out because defects are not color related and aircrafts are painted in different colors, depending on the company. Images of the datasets can be observed in Figure 2.

Following the above procedure, two datasets were created, one containing images from each category of the defects described above and one contains images with and without defects. The second dataset in the no-defect category has images of screws, gaps, small plates etc., objects that the classifier will need to distinguish from the defects. Figure 2 shows images from the two datasets with and without defects.

The defect/no defect dataset, which we will refer as binary for simplicity, contains 1059 images, 576 of defects and 483 of non-defects. The other dataset, referred as the defect dataset, contains 576 images of the 7 types of defects. Both datasets were relatively small but gathering images was very challenging under the current circumstances (COVID-19 restrictions, flights reductions etc.). To try to overcome this, we carried out a custom split of the images between training, and validation, with 88% for training, 9% for validation and the rest for testing for both datasets. This was carried out to give the opportunity to the classifiers to learn as much as possible from the dataset. For the binary dataset, the splitting can be observed in Table 1 and for the defect dataset in Table 2.

## 3. Defect Classification Algorithms

As previously mentioned, one of the challenges of the classification problems in applications in aerospace is the small amount of data available. In this paper, we tried to address this by proposing a two-step classification approach with a combination of different classifiers. In the first step, a classifier decides if the image contains a defect and if this is true in the second step, the defect is classified by a different classifier. The classifiers are a combination of pretrained CNNs on ImageNet [29], which we retrained with the use of transfer learning [30]. In the first step, a DenseNet201 model is used and in the second, an ensemble of different CNNs as can be observed in Figure 3.

Transfer learning refers to a technique of retraining a CNN that has already been trained in very large dataset, such as Imagenet [29]. Even though the dataset that the CNN is been initially trained in is irrelevant to the problem research, ref. [30] has shown that the benefits of this technique are substantial. There are mainly two approaches on how to implement transfer learning; in the first, only the convolutional layers of the trained network are used as feature extractors [31] and then the features are fed to a different classifier, such as support vector machines [31]. In the second approach, which is used in this paper, the head of the neural network (fully connected layers) is replaced. The output of the new connected layers will match the number of the categories of our classifier. The new neural network is initially trained by keeping all the weights of the convolution layers frozen/non trainable. Then, to fine tune the model, a number of the layers are unfrozen and the training of the network is updated. The basic rule for unfreezing layers is, the less the data, the less layers to unfreeze. In addition, because the initial/bottom layers of a CNN extract more abstract features that can be used in any type of image, we unfreeze (for training) the layers closer to the top of the network. Another point that needs attention during both training rounds is not to update the weights of the batch normalization layers. These layers contain two non-trainable weights, tracking the mean and variance of the inputs that usually get updated during training. So, if we unfreeze these layers during fine tuning, the updates applied will destroy what the model has learned.

The models were implemented using TensorFlow [32], as this is a well-established deep learning library, widely used for both commercial applications and research. Because TensorFlow contains around forty pre-trained networks, we needed to identify those that fit better on our datasets. To achieve this, we trained each network for five epochs with the convolutional layers frozen. To continue with fine tuning, we chose the best four pretrained networks for each classifier. For the binary classifier, the models that performed better were Mobilenet, DenseNet201, ResNet15V2 and InceptionResNetV2. For the defect classifier, the four models with the best results were EfficientNetB1, EfficientNetB5, EfficientNetB4 and DenseNet169.

To improve the performance of the chosen models, we fine-tuned them for another ten epochs. For fine-tuning, we unfroze the last 10% of the layers of each model and reduced the learning rate by a factor of ten compared to the initial one. In addition, techniques of reduce learning and early stopping were used. Both techniques are included in TensorFlow libraries. In the reduce learning technique, the learning rate of the optimizer is reduced if the validation loss has not improved for a certain number of epochs. Similar in the early stopping as the name suggests, training stops if our metric (in this case, validation loss) has not improved for a certain number of epochs and the graph with the best weights is saved. Both techniques were used to prevent overfitting.

In addition to the CNN, a random forest was trained. The initial idea was to use in the first step both the CNN and the random forest but the overall benefit of this was low. For training, the random forest we have extracted the features of Hu moments, color histogram and Haralick. The overall accuracy of the random forest classifier was 76%.

## 4. Results

As discussed in the previous chapter, the initial training of five epochs has been carried out for each of the pretrained models of TensorFlow. The results of the four best networks for the defect recognition can be observed in Table 3.

The results of the best four networks for the defect classification can be observed in Table 4.

As expected, due to the small number of images and due to the lack of fine tuning especially for the defect classifier, the accuracy in both the validation and testing images was relatively low. At this stage, no further analysis was carried out or any extra metrics, such as confusion matrices or classification reports, as the purpose was to identify the best CNNs for each of the datasets.

After this initial training, each of the networks were further trained, as discussed for another ten epochs. The results of the training can be observed in Table 5.

The same procedure was followed for the other set of classifiers for the defect classification. The results can be observed in Table 6.

To understand better the behavior of the CNNs while trained, the validation loss was taken into account. This metric, together with the validation accuracy, can illustrate when the CNN will start overfitting. Usually, when the validation loss does not improve, but the validation accuracy does, overfitting occurs. This is also the main reason why we used the techniques of reduce learning and early stopping.

To decide which of the above eight CNNs to use in the proposed system, further metrics were produced. For each of the models, a classification report and a confusion matrix was produced to measure the performance in the test data. A classification report measures the values of precision, recall and F1-score [33]. Precision quantifies the number of correct positive predictions. It is defined as the ratio of true positives divided by the sum of true positives and false positives [33]. It shows how precise/accurate the model is. It is very useful if the false positive cost is high, which in our case was not. If one misclassifies a non-defect, it will produce an extra load of work for the inspector but it is not critical. Recall is the ratio of correctly predicted positive predictions against all the predictions in the actual class [33]. It is the ratio of true positives divided by the sum of true positives and false negatives. In simple terms, recall shows how many of the predictions in the class are actual positives. It is the metric we can use if there is a high cost of false negatives; in our case, if we misclassify a defect as non-defect. The F1 score is calculated as the multiplication of precision and recall, divided by the sum of precision and recall and then multiplied by 2 [33]. The F1 score can be interpreted as the harmonic mean of both precision and recall. The F1 score can also be interpreted as the average of precision and recall. It is a very valuable metric, especially when both errors caused by false positives and false negatives are undesirable.

Taking into consideration all the above, we created a classification report with the above metrics for each of the models.

In Table 7, the combined classification reports can be observed for all four models for defect recognition and in Table 8, the combined confusion matrices.

From the above tables, we can observe that DenseNet201 performs very well with high precision. The results from the confusion matrix show that the model has predicted correct eighteen out of the twenty-two images containing a defect and nine out of eleven images for the no defect category.

Comparing InceptionResNetV2 and DenseNet201, we can observe that the first has a better precision than DenseNet201 for the defect category by its recall value being much lower. This is also reflected in the confusion matrix, where InceptionResNetV2 has more false negatives. In addition, the F1 score for DenseNet201 is higher in both categories. Because misclassifying a defect is critical in our application, we can state that DenseNet201 performs better.

From the above results, in can be observed that DenseNet201 has the best overall accuracy with 81.82%, the best precision and recall values for the defect class. In addition, it has the least false negatives and the best F1 score for both classes. Another test we performed was to combine the classifiers in an ensemble to investigate whether any improvements in the metrics were possible. The ensemble of classifiers did not give better results, compared to DenseNet201.

The same procedure was followed for the defect classification models and the results of the metrics and confusion matrices can be observed in Table 9, Table 10, Table 11, Table 12, Table 13, Table 14, Table 15 and Table 16.

From the above matrices, the performance of the models for the defect classification is relatively low. However, this is due to the number of images in the dataset and because the dataset was unbalanced. To improve performance and ensure the predictions are more consistent, we used the ensemble model. We combined all four models to create a new model in which the input image is fed into all four models. The predictions of each of the models are passed to a layer that is added at the end of the model. This final layer averages the predictions of the four models and returns array with the new values. This technique, especially in our case where the performance of the models is similar, provides a more consistent outcome for all the different classes. The results for the ensemble model can be observed in Table 17 and Table 18.

For the ensemble model, although in some categories it may have worse performance than others, its overall performance is better. It has positive predictions for all the categories in comparison with other models and its overall accuracy is above the average value of the models.

Finally, we tested the whole pipeline of our algorithm. We first fed the test images to the defect recognition model and then, if the image had a defect, we passed it to the defect classifier. As a defect recognition model, we have chosen the DenseNet201 and for the defect classification, the ensemble model. As we have used the same test dataset, the results of the defect recognition model are the same as Table 7 and Table 8 and for the ensemble, similar to the Table 17 and Table 18. However, by filtering through the first step, the images that we achieved 100% accuracy for were the categories of the missing or damaged exterior paint and primer and dents.

Although the results are promising, the overall accuracy of the defect classifier is low. As previously mentioned, this is mainly due to the small number of images and because the dataset is very unbalanced. Taking into consideration the accuracy for the defect recognition classifier together with the number of images, we believe that by having around five hundred images for each defect category, we will be able to improve significantly not only the performance of the defect classifier but also of the overall process.

## 5. Conclusions

In this paper, we have presented the development of a two-step process for defect recognition and classification of aircraft structures. A dataset was created from real aircraft defects taken in TUI’s maintenance hangar. On the one hand, the lack of defects on aircrafts made the creation of the dataset very challenging and on the other, the recognition of defects is crucial for the safety of the passengers and crew. To overcome this, we proposed a two-step process method. Firstly, we recognized the defect and then we classified it. This method has the advantage of using two different classifiers, one for defect recognition and one for defect classification. By splitting the process of defect recognition and classification in two, we improved the accuracy. This is because first, we can train the defect recognition model with more data, thus making it more accurate. In addition, in this first step, we perform with higher accuracy the most significant part of finding the defect. Secondly, we use a dedicated classifier for defect classification. This gives the opportunity to the second classifier to learn more effectively the differences between the different types of defects, as it does not have to learn any of the non-defect images.

The results of the first step had an accuracy **81.82%**, which is quite high considering the small training dataset. In the second step, for the defects of missing or damaged exterior paint and primer and dents, we achieved 100% accuracy.

Although the results are promising, future work will be carried out in increasing the defect dataset, especially in adding more images in the very small categories to improve the unbalanced dataset. In addition, the process will be combined with a UAV inspection for real time recognition and classification

## Figures and Tables

**Figure 1 sensors-22-04682-f001:**
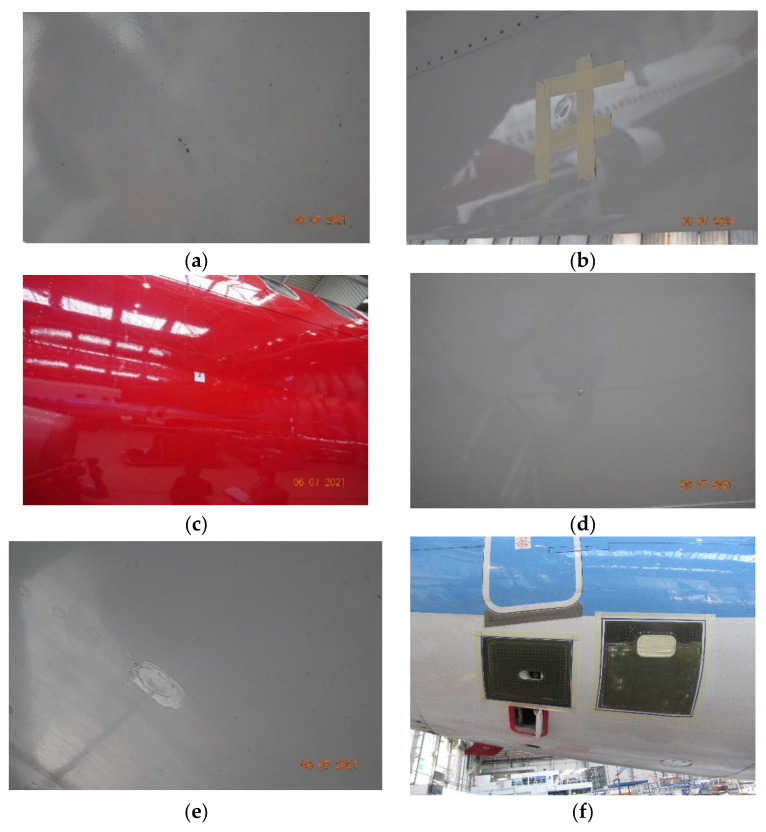
Images of different types of defects in aircraft structures. (**a**) Missing paint, (**b**) dents, (**c**) lighting strike damage, (**d**) lighting strike fastener repair, (**e**) blend/rework repair (material removed and then re-protected with exterior paint); (**f**) double patch repair.

**Figure 2 sensors-22-04682-f002:**
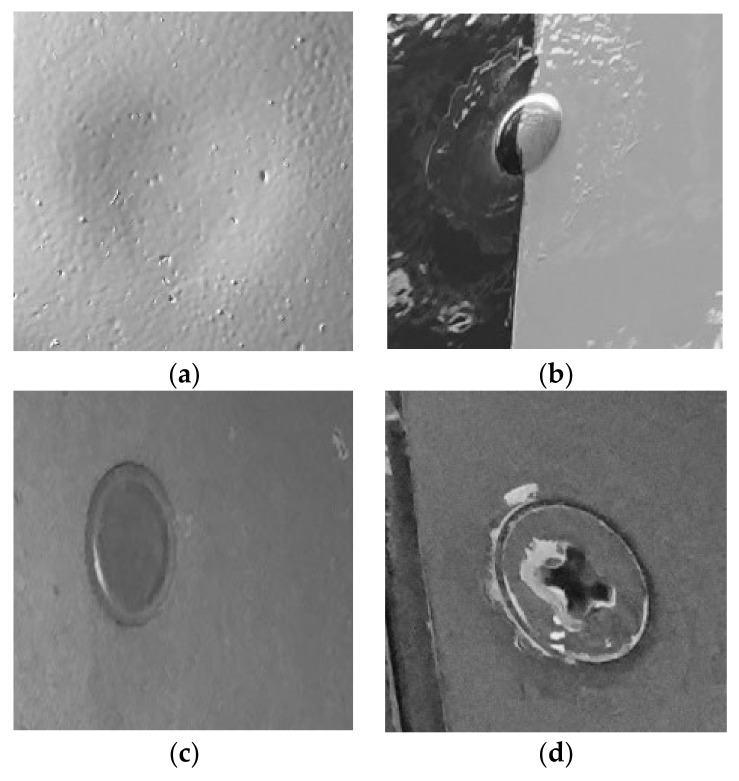
Sample images from the two datasets created for training the classifiers. (**a**) An image of a dent, (**b**) a lighting strike fastener repair; (**c**,**d**) are images with objects that are not defects.

**Figure 3 sensors-22-04682-f003:**
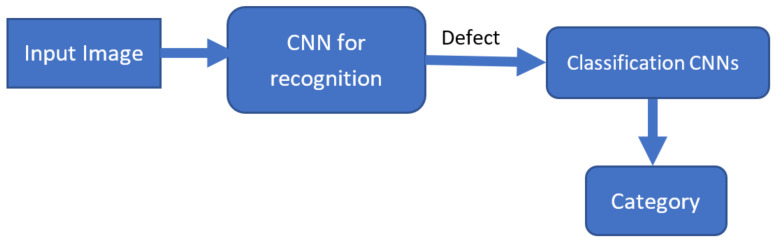
Block diagram of the two-step process for defect recognition and classification.

**Table 1 sensors-22-04682-t001:** Dataset split for training, validating and testing the defect/non defect classifier.

Dataset Split	Non-Defect	Defect
Training	426	576
Validation	46	63
Testing	11	22

**Table 2 sensors-22-04682-t002:** Dataset split for training, validating and testing the defect classifier.

Dataset Categories	Training	Validation	Testing
Missing or Damaged Exterior Paint and Primer	77	8	3
Dents	151	25	6
Reinforcing Patch Repairs	109	10	4
Nicks, Scratches and Gouges	57	6	3
Blend/Rework Repairs	82	10	3
Lighting Strike Damage	4	1	1
Lighting Strike Fastener Repairs	11	3	2

**Table 3 sensors-22-04682-t003:** Performance of the 4 best out of 40 pretrained networks for the binary classifier after 5 epochs.

Model	Validation Accuracy	Testing Accuracy
Mobilenet	0.80	0.63
DenseNet201	0.84	0.81
ResNet152V2	0.74	0.88
InceptionResNetV2	0.79	0.85

**Table 4 sensors-22-04682-t004:** Performance of the 4 best out of 40 pretrained networks for the defect classifier after 5 epochs.

Model	Validation Accuracy	Testing Accuracy
EfficientNetB1	0.60	0.68
EfficientNetB5	0.63	0.68
EfficientNetB4	0.71	0.63
DenseNet169	0.70	0.60

**Table 5 sensors-22-04682-t005:** Performance of the 4 best pretrained networks for binary classifier after fine tuning for a total of 15 epochs.

Model	Validation Loss	Validation Accuracy	Testing Accuracy
MobileNet	0.39	0.79	0.63
DenseNet201	0.46	0.84	0.82
InceptionResNetV2	0.43	0.77	0.69
ResNet152V2	0.61	0.78	0.66

**Table 6 sensors-22-04682-t006:** Performance of the 4 best pretrained networks for defect classifier after fine tuning for a total of 15 epochs.

Model	Validation Loss	Validation Accuracy	Testing Accuracy
EfficientNetB1	0.76	0.66	0.72
EfficientNetB5	0.52	0.85	0.82
EfficientNetB4	0.54	0.79	0.72
DenseNet169	0.82	0.71	0.82

**Table 7 sensors-22-04682-t007:** Combined classification reports for defect recognition classifiers.

MobileNet				
	Precision	Recall	F1 Score	Sum of Images
Defect	0.83	0.68	0.75	22
No Defect	0.53	0.72	0.61	11
Accuracy				**69.70%**
**ResNet15V2**				
	**Precision**	**Recall**	**F1 Score**	**Sum of Images**
Defect	0.88	0.68	0.76	22
No Defect	0.56	0.81	0.66	11
Accuracy				**72.73%**
**InceptionResNetV2**				
	**Precision**	**Recall**	**F1 Score**	**Sum of Images**
Defect	0.93	0.68	0.78	22
No Defect	0.58	0.90	0.71	11
Accuracy				**75.76%**
**DenseNet201**				
	**Precision**	**Recall**	**F1 Score**	**Sum of Images**
Defect	0.9	0.82	0.85	22
No Defect	0.69	0.82	0.75	11
Accuracy				**81.82%**

**Table 8 sensors-22-04682-t008:** Combined confusion matrices for defect recognition classifiers.

	MobileNet	
Actual	Predicted Class	Predicted Class
	Defect	No Defect
Defect	15	7
No Defect	3	8
	**ResNet15V2**	
**Actual**	**Predicted Class**	**Predicted Class**
	Defect	No Defect
Defect	15	7
No Defect	2	9
	**InceptionResNetV2**	
**Actual**	**Predicted Class**	**Predicted Class**
	Defect	No Defect
Defect	15	7
No Defect	1	10
**Actual**	**Predicted Class**	**Predicted Class**
	Defect	No Defect
Defect	18	4
No Defect	2	9

**Table 9 sensors-22-04682-t009:** Classification report of Dense169 for defect recognition.

Dense169				
	Precision	Recall	F1 Score	Sum of Images
Missing or Damaged Exterior Paint and Primer	0.22	0.66	0.33	3
Dents	0.67	0.33	0.44	6
Reinforcing Patch Repairs	1	0.5	0.66	4
Nicks, Scratches and Gouges	1	0.33	0.5	3
Blend/Rework Repairs	0.5	0.66	0.57	3
Lighting Strike Damage	1	1	1	1
Lighting Strike Fast Repairs	1	1	1	2
Accuracy				**54.55%**

**Table 10 sensors-22-04682-t010:** Confusion natrix for Dense 169.

Actual	Predicted Class						
	Missing/Damaged Exterior Paint and Primer	Dents	Reinforcing Patch Repairs	Nicks, Scratches and Gouges	Blend/Rework Repairs	Lighting Strike	Lighting Strike Fast Repairs
Missing/Damaged Paint and Primer	2	1	0	0	0	0	0
Dents	3	2	0	0	1	0	0
Reinforcing Patch Repairs	2	0	2	0	0	0	0
Nicks, Scratches and Gouges	1	0	0	1	1	0	0
Blend/Rework Repairs	1	0	0	0	2	0	0
Lighting Strike	0	0	0	0	0	1	0
Lighting Strike Fast Repairs	0	0	0	0	0	0	2

**Table 11 sensors-22-04682-t011:** Classification report of EfficientNetB1 for defect classification.

EfficientNetB1				
	Precision	Recall	F1 Score	Sum of Images
Missing or Damaged Exterior Paint and Primer	0.6	1	0.75	3
Dents	1	1	1	6
Reinforcing Patch Repairs	0.5	0.5	0.5	4
Nicks, Scratches and Gouges	0	0	0	3
Blend/Rework Repairs	0.66	0.66	0.66	3
Lighting Strike Damage	1	1	1	1
Lighting Strike Fast Repairs	1	1	1	2
Accuracy				**72.73%**

**Table 12 sensors-22-04682-t012:** Confusion matrix of EfficientNetB1.

Actual	Predicted Class						
	Missing/Damaged Exterior Paint and Primer	Dents	Reinforcing Patch Repairs	Nicks, Scratches and Gouges	Blend/Rework Repairs	Lighting Strike	Lighting Strike Fast Repairs
Missing or Damaged Exterior Paint and Primer	3	0	0	0	0	0	0
Dents	0	6	0	0	0	0	0
Reinforcing Patch Repairs	1	0	2	1	0	0	0
Nicks, Scratches and Gouges	1	0	1	0	1	0	0
Blend/Rework Repairs	0	0	1	0	2	0	0
Lighting Strike Damage	0	0	0	0	0	1	0
Lighting Strike Fast Repairs	0	0	0	0	0	0	2

**Table 13 sensors-22-04682-t013:** Classification report of EfficientNetB4 for defect classification.

EfficientNetB4				
	Precision	Recall	F1 Score	Sum of Images
Missing or Damaged Exterior Paint and Primer	0.5	1	0.66	3
Dents	0.83	0.83	0.83	6
Reinforcing Patch Repairs	0.5	0.5	0.5	4
Nicks, Scratches and Gouges	1	0.33	0.5	3
Blend/Rework Repairs	0	0	0	3
Lighting Strike Damage	1	1	1	1
Lighting Strike Fast Repairs	1	1	1	2
Accuracy				**63.64%**

**Table 14 sensors-22-04682-t014:** Confusion matrix of EfficientNetB4.

Actual	Predicted Class						
	Missing/Damaged Exterior Paint and Primer	Dents	Reinforcing Patch Repairs	Nicks, Scratches and Gouges	Blend/Rework Repairs	Lighting Strike	Lighting Strike Fast Repairs
Missing or Damaged Exterior Paint and Primer	3	0	0	0	0	0	0
Dents	1	5	0	0	0	0	0
Reinforcing Patch Repairs	0	1	2	0	1	0	0
Nicks, Scratches and Gouges	1	0	0	1	1	0	0
Blend/Rework Repairs	1	0	2	0	0	0	0
Lighting Strike Damage	0	0	0	0	0	1	0
Lighting Strike Fast Repairs	0	0	0	0	0	0	2

**Table 15 sensors-22-04682-t015:** Classification report of EfficientNetB5 for defect classification.

EfficientNetB5				
	Precision	Recall	F1 Score	Sum of Images
Missing or Damaged Exterior Paint and Primer	1	1	1	3
Dents	1	0.83	0.90	6
Reinforcing Patch Repairs	0.16	0.25	0.2	4
Nicks, Scratches and Gouges	1	0.66	0.8	3
Blend/Rework Repairs	0	0	0	3
Lighting Strike Damage	1	1	1	1
Lighting Strike Fast Repairs	1	1	1	2
Accuracy				**63.64%**

**Table 16 sensors-22-04682-t016:** Confusion matrix of EfficientNetB5.

Actual	Predicted Class						
	Missing/Damaged Exterior Paint and Primer	Dents	Reinforcing Patch Repairs	Nicks, Scratches and Gouges	Blend/Rework Repairs	Lighting Strike	Lighting Strike Fast Repairs
Missing or Damaged Exterior Paint and Primer	3	0	0	0	0	0	0
Dents	0	5	1	0	0	0	0
Reinforcing Patch Repairs	0	0	1	0	3	0	0
Nicks, Scratches and Gouges	0	0	1	2	0	0	0
Blend/Rework Repairs	0	0	3	0	0	0	0
Lighting Strike Damage	0	0	0	0	0	1	0
Lighting Strike Fast Repairs	0	0	0	0	0	0	2

**Table 17 sensors-22-04682-t017:** Classification report of the ensemble model for defect classification.

Ensemble				
	Precision	Recall	F1 Score	Sum of Images
Missing or Damaged Exterior Paint and Primer	0.6	1	0.75	3
Dents	1	0.83	0.90	6
Reinforcing Patch Repairs	0.5	0.5	0.5	4
Nicks, Scratches and Gouges	0.5	0.33	0.4	3
Blend/Rework Repairs	0.33	0.33	0.33	3
Lighting Strike Damage	1	1	1	1
Lighting Strike Fast Repairs	1	1	1	2
Accuracy				**68.18%**

**Table 18 sensors-22-04682-t018:** Confusion matrix of the Ensemble.

Actual	Predicted Class						
	Missing/Damaged Exterior Paint and Primer	Dents	Reinforcing Patch Repairs	Nicks, Scratches and Gouges	Blend/Rework Repairs	Lighting Strike	Lighting Strike Fast Repairs
Missing or Damaged Exterior Paint and Primer	3	0	0	0	0	0	0
Dents	1	5	0	0	0	0	0
Reinforcing Patch Repairs	0	0	2	1	1	0	0
Nicks, Scratches and Gouges	1	0	0	1	1	0	0
Blend/Rework Repairs	0	0	2	0	1	0	0
Lighting Strike Damage	0	0	0	0	0	1	0
Lighting Strike Fast Repairs	0	0	0	0	0	0	2

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
