# Peer review of "Defects Recognition Algorithm Development from Visual UAV Inspections"

_sensors, 2022, doi:10.3390/s22134682_

Round 1

Reviewer 1 Report

In this paper the authors presented a two-step process for automating the defect recognition and classification from visual images acquired using a UAV on aircraft structures. Overall, the paper is well written and falls within the scope of the journal. I suggest the following minor changes for the paper to be accepted for publication:
-    Please specify the categories mentioned in the abstract (Line 23)
-    Please be consistent on the terms used in your manuscript (i.e. UAVs or UAV's, 2step or twostep or two-step, etc)
-    Correct title of Section 3
-    Authors mention (Line 310) that the performance of the models for the defect classification is relatively low and that this is due to the number of images in the dataset. Please comment further on what number of images would be ideal to have in order to improve the performance of models
-    In Conclusions section, please name the defects that achieved 100% accuracy. Furthermore, please suggest future work in such interesting research topic

Author Response

In this paper the authors presented a two-step process for automating the defect recognition and classification from visual images acquired using a UAV on aircraft structures. Overall, the paper is well written and falls within the scope of the journal.

Reply: Thank you for the comments and for taking the time to review our paper. Please see below reply to your comments.

I suggest the following minor changes for the paper to be accepted for publication:

Please specify the categories mentioned in the abstract (Line 23): Completed

Please be consistent on the terms used in your manuscript (i.e. UAVs or UAV's, 2step or twostep or two-step, etc): Checked and updated accordingly

Correct title of Section 3: Done

Authors mention (Line 310) that the performance of the models for the defect classification is relatively low and that this is due to the number of images in the dataset. Please comment further on what number of images would be ideal to have in order to improve the performance of models: Completed in the document

In Conclusions section, please name the defects that achieved 100% accuracy: Done

Furthermore, please suggest future work in such interesting research topic: This is updated accordingly in the document

Reviewer 2 Report

In the article the authors presented the development of a two-step process for defect recognition and classification of aircraft structures. First, the defect was recognized and then classified by its type with the use of Convolutional Neural Network. A dataset was created from real aircraft defects grouped into seven types.

I recommend taking into account following suggestions:

1. A thorough proofreading is necessary. Some expressions are to colloquial. 'So' is overused. Some sentences are hard to understand, comas are nedeed.

2. Present continuous is (e.g. line 111) is not appropriate in the body of the article

3. Choose one version: two-step, 2-step, twostep; UAV's, UAVs; f1-score, F1 Score.

4. What are 'some categories' in the abstract (line 23), please specify.

5. Check the title of section 3.

6. Digits in the text should be substituted by numbers written in words (e.g. lines 184, 207, 2011 - '4' written as 'four').

7. The UAV abbreviation should be expanded (one time).

8. Caption under the figure 2 should start with 'sample images'.

9. What is the reason reason for calculating f1 score in the proposed way?

10. Where exactly in your research sensors appear? Of what kind?

11. Reference [31] - more convenient would be: Abadi, M; et al. instead of forty names.

12. In conclusions it would be worth to mention which types of defects were classified with 100% accuracy.

Author Response

In the article the authors presented the development of a two-step process for defect recognition and classification of aircraft structures. First, the defect was recognized and then classified by its type with the use of Convolutional Neural Network. A dataset was created from real aircraft defects grouped into seven types.

I recommend taking into account following suggestions:

A thorough proofreading is necessary. Some expressions are to colloquial. 'So' is overused. Some sentences are hard to understand, comas are needed: Updated accordingly

Present continuous is (e.g. line 111) is not appropriate in the body of the article: Changed

Choose one version: two-step, 2-step, twostep; UAV's, UAVs; f1-score, F1 Score: Updated accordingly

What are 'some categories' in the abstract (line 23), please specify: Specified

Check the title of section 3: Title updated

Digits in the text should be substituted by numbers written in words (e.g. lines 184, 207, 2011 - '4' written as 'four'): Done

The UAV abbreviation should be expanded (one time): Done

Caption under the figure 2 should start with 'sample images': Completed

What is the reason reason for calculating f1 score in the proposed way?: Done, updated in the document

Where exactly in your research sensors appear? Of what kind?: Added also in the abstract, line 19 Line 75: Added The imaging sensor used was a Sony ....

Reference [31] - more convenient would be: Abadi, M; et al. instead of forty names: Updated according to the comment by the reviewer

In conclusions it would be worth to mention which types of defects were classified with 100% accuracy: Updated and mentioned in the conclusions as suggested by the reviewer

Reviewer 3 Report

The manuscript is written with clear understanding of the project addressed. However, there are major concerns that need to be addressed to enhance the quality of the manuscript. My specific comments are as follows:

Abstract:

Elaborate more on the methods used in this study.

Introduction:

Page2Line47: “Several attempts have been made and most of them can be divided in two categories: the ones using more traditional image processing techniques [5, 16, 17, 18] and the ones using machine learning [19, 20, 21, 22, 23, 24].…” Discuss several attempts by those literatures

Highlight the utilization of CNN

Based on your objectives, please compare how your study is different from those that have already been published

Dataset:

Caption of Figure 2: (c) missing??

How about the sample dataset/preparation? Explain briefly

Is there any image preprocessing conducted for each image?

P4L127: “To try to overcome this we did a custom split of images between training, validation and testing for both datasets.” Specify the split ratio of these datasets

Add data analysis part

“Defect Classific12546005ation Algorithms”: spelling error

P5L145: “The classifiers are a combination of CNNs.” What type of CNN models?

P5L164: “There are two approaches on how to implement transfer learning, in the first only the Convolutional layers of the trained network are used as feature extractors…” Add citation

P5L167: “In the second approach which is used in this paper the head of the neural network (Fully connected layers) is replaced with new connected layers so the number of outputs..” Add citation

Results:

“Table 3: Performance of the 4 best pretrained networks for the binary classifier after 6 epochs” Should be 5 epochs right? The same as Table 4

Please specify which tables (Table 5 & 6) denote for binary classifier or defect classifier?

P7L235: “It is defined as the ratio of True Positives divided by the sum of True Positives and False Positives.” Add citation

P7L240: “It is the ratio of True Positive divided by the sum of True Positives and False Negatives.” Add citation

Please combine the classification report of CNN models for defect recognition in one single table for comparison.

Do the same for confusion matrix of CNN models.

Which one produce the best results?

Instead of mentioning the results, the authors should justify/explain the findings

Conclusions:

Justify the main finding of your study.

Add recommendation for future studies.

General comments:

Please check the reference styles and grammar of the manuscript.

Author Response

The manuscript is written with clear understanding of the project addressed. However, there are major concerns that need to be addressed to enhance the quality of the manuscript. My specific comments are as follows:

Abstract:

Elaborate more on the methods used in this study: Added further explanation from Line 23 to Line 30

Introduction:

Page2Line47: “Several attempts have been made and most of them can be divided in two categories: the ones using more traditional image processing techniques [5, 16, 17, 18] and the ones using machine learning [19, 20, 21, 22, 23, 24].…” Discuss several attempts by those literatures: Done

Line 57 to 70

“Several attempts have been made and most of them can be divided in two categories: the ones using more traditional image processing techniques [5, 16, 17, 18] and the ones using machine learning [19, 20, 21, 22, 23, 24]. In the first category image features like convexity or signal intensity [5] are used. In [18] the propose a method using histogram comparisons or structural similarity. In addition, in [16] and [17] they propose the use of Neural Networks trained on feature vectors extracted from Contourlet transform. These techniques have very good accuracy in the test data but are failing to generalize well and need continuous tuning. On the other hand, algorithms using Convolutional Neural Networks (CNN) have showed good results in defect detection [19, 20, 21, 25]. In [19] and [20] CNNs are used as feature extractors and then either a Single Shot Multibox Detector (SSD) or a Support Vector Machine (SVM) are used for the classification. Also, the use of Faster CNN is proposed for classification and localization [22]. In addition, the use of UAVs together with deep learning algorithms is proposed for tagging and localization of concrete cracks [26,27].”

Highlight the utilization of CNN: Done Line 64 to Line 70

 “On the other hand, algorithms using Convolutional Neural Networks (CNN) have showed good results in defect detection [19, 20, 21, 25]. In [19] and [20] CNNs are used as feature extractors and then either a Single Shot Multibox Detector (SSD) or a Support Vector Machine (SVM) are used for the classification. Also, the use of Faster CNN is proposed for classification and localization [22]. In addition, the use of UAVs together with deep learning algorithms is proposed for tagging and localization of concrete cracks [26,27].”

Based on your objectives, please compare how your study is different from those that have already been published.

 Done.  Line 79 to Line 84

“In this paper we propose a two-step classification process of an ensemble of machine learning classifiers for both defect recognition and classification. In this two-step process we are using pretrained CNNs to both recognize and classify a series of defects in aircraft metallic and composite structures. In the first step we are performing the defect recognition and in the second step the defect classification. By combining the results of different classifiers, we can address better the issue of small datasets and produce results with accuracy reaching 82% in the defect recognition step.

Dataset:

Caption of Figure 2: (c) missing??: Done

How about the sample dataset/preparation? Explain briefly: Completed

Is there any image preprocessing conducted for each image?: Done. The images were preprocessed and used to train different pretrained CNNs with the use of transfer learning.

P4L127: “To try to overcome this we did a custom split of images between training, validation and testing for both datasets.” Specify the split ratio of these datasets: Done

Add data analysis part: Added a brief image pre-processing stage and the way the dataset was split for the training. As the image is fed directly to the CNN as is (Not any further feature extraction is needed) nothing further is needed.

“Defect Classific12546005ation Algorithms”: spelling error: Completed

P5L145: “The classifiers are a combination of CNNs.” What type of CNN models?: Done

P5L164: “There are two approaches on how to implement transfer learning, in the first only the Convolutional layers of the trained network are used as feature extractors…” Add citation: Done

P5L167: “In the second approach which is used in this paper the head of the neural network (Fully connected layers) is replaced with new connected layers so the number of outputs..” Add citation: Done

Results:

“Table 3: Performance of the 4 best pretrained networks for the binary classifier after 6 epochs” Should be 5 epochs right? The same as Table 4: Done - updated

Please specify which tables (Table 5 & 6) denote for binary classifier or defect classifier?: Done

P7L235: “It is defined as the ratio of True Positives divided by the sum of True Positives and False Positives.” Add citation: Done

P7L240: “It is the ratio of True Positive divided by the sum of True Positives and False Negatives.” Add citation: Done

Please combine the classification report of CNN models for defect recognition in one single table for comparison: Completed

Do the same for confusion matrix of CNN models: Completed

Which one produce the best results?: Completed in the document

Instead of mentioning the results, the authors should justify/explain the findings: Updated - completed in the document

Conclusions:

Justify the main finding of your study: Done

Add recommendation for future studies: Done

General comments:

Please check the reference styles and grammar of the manuscript: Completed